# Progress in Multisensory Synergistic Salt Reduction

**DOI:** 10.3390/foods13111659

**Published:** 2024-05-25

**Authors:** Shujing Liu, Yuxiang Gu, Ruiyi Zheng, Baoguo Sun, Lili Zhang, Yuyu Zhang

**Affiliations:** 1Food Laboratory of Zhongyuan, Beijing Technology and Business University, Beijing 100048, China; 19544541895@163.com (S.L.); yuxiang.gu@outlook.com (Y.G.); yolo17807640109@163.com (R.Z.); zhanglili921116@163.com (L.Z.); 2Key Laboratory of Geriatric Nutrition and Health, Beijing Technology and Business University, Ministry of Education, Beijing 100048, China; sunbg@btbu.edu.cn; 3Key Laboratory of Flavor Science of China General Chamber of Commerce, Beijing Technology and Business University, Beijing 100048, China

**Keywords:** salt reduction, salty taste perception, multisensory interaction, flavor, nutrition

## Abstract

Excessive salt intake, primarily from sodium chloride prevalent in modern food processing, poses a significant public health risk associated with hypertension, cardiovascular diseases and stroke. Researchers worldwide are exploring approaches to reduce salt consumption without compromising food flavor. One promising method is to enhance salty taste perception using multisensory synergies, leveraging gustatory, olfactory, auditory, visual, tactile and trigeminal senses to decrease salt intake while preserving food taste. This review provides a comprehensive overview of salt usage in foods, mechanisms of salty taste perception and evaluation methods for saltiness. Various strategies for reducing salt consumption while maintaining food flavor are examined, with existing salt reduction methods’ advantages and limitations being critically analyzed. A particular emphasis is placed on exploring the mechanisms and potential of multisensory synergy in salt reduction. Taste interactions, olfactory cues, auditory stimulation, visual appearance and tactile sensations in enhancing saltiness perception are discussed, offering insights into developing nutritious, appealing low-sodium foods. Furthermore, challenges in current research are highlighted, and future directions for effective salt reduction strategies to promote public health are proposed. This review aims to establish a scientific foundation for creating healthier, flavorful low-sodium food options that meet consumer preferences and wellness needs.

## 1. Introduction

Salt, regarded as the cornerstone of the modern food industry and human dietary culture and known as the “king of all flavors” [1]. It provides a salty taste (one of the five basic tastes), enhances umami, highlights sweetness, balances sourness, inhibits bitterness, contributes to good texture performance and extends the shelf life of food [2]. Its dissociation into sodium and chloride ions is essential for maintaining vital life activities and metabolism processes. However, with the popularity of highly processed foods and the excessive pursuit of salt in cooking, salt intake (10.78 g per day per capita globally) is far beyond physiological needs (no more than 5 g of salt per day, as recommended by the World Health Organization (WHO)) [2,3,4,5]. A high-salt diet can cause changes in osmotic pressure and an increase in extracellular fluid, leading to sodium and water retention. These changes can elevate blood pressure, which in turn can lead to cardiovascular diseases such as hypertension and stroke [6]. High-salt diet has been listed as one of the major dietary risk factors by the Global Burden of Disease [1,4].

In response to this challenge, researchers worldwide are actively exploring effective ways to reduce salt intake. Despite various strategies for salt reduction, such as salt substitutes, changing the shape and structure of salt and innovation in food processing technology [7], these methods often have limited applicability, sustainability or high costs, failing to meet the public’s urgent demand for a healthy diet [8,9]. Therefore, a salt reduction approach based on multisensory synergistic effects has emerged. This method aims to decrease salt intake without compromising food flavor by leveraging interactions between taste and other senses, such as vision, hearing and smell. This innovative approach not only satisfies consumers’ desires for both health and culinary enjoyment but also offers novel insights and theoretical support for producing high-quality, low-sodium foods [10,11,12].

This paper provides an overview of the current status of salt content, salty taste perception mechanisms and salt evaluation systems in food, and delves into strategies for reducing salt intake while preserving food flavor. By analyzing the advantages and limitations of existing salt reduction methods, this paper concentrates on the mechanisms and future development prospects of multisensory synergistic salt reduction. Finally, the paper outlines the challenges and future directions of current research, aiming to provide scientific evidence and technical support for the development of low-sodium healthy food products, reducing the dietary salt intake of the population and promoting a healthy diet for all, thereby contributing to the health and well-being of mankind.

## 2. Salty Taste Perception Mechanism and Sodium Salt Reduction Strategies

### 2.1. Current Status of Salt Content in Food and Hazards of Excessive Salt Intake

Table salt, predominantly consisting of sodium chloride (NaCl), serves as an indispensable ingredient in the modern food industry. Its ubiquitous application in food processing and cooking renders it the primary source of sodium intake in the human diet, comprising approximately 90% [13]. The sodium content label on the food packaging can be converted to reflect the salt content of the food. Both processed food and natural ingredients, such as frozen shrimp, ham, sausages, bread and processed snacks [14,15], rely on salt to enhance flavor, improve texture and inhibit the growth of harmful microorganisms. Nonetheless, the actual quantity of salt employed (on a global scale, with the average daily salt intake for adults reaching as high as 10.78 g/day) far surpasses the WHO’s recommended daily salt intake limit of no more than 5 g [16,17]. Specifically, the average daily intake in China is 9.3 g, in the United States it is 8.7 g, and in European countries, it ranges between 8 and 12 g [1].

This pattern of excessive consumption is particularly prevalent in prepackaged foods and artisanal products. A study analyzing data from 4082 commercial food products unveiled an average sodium content of 1018.6 mg/100 g, with high-sodium foods (defined as those with sodium content ≥600 mg/100 g) constituting 37.30% of the total, particularly notable in seasonings, soy products, meat and egg items, with sodium content ranging from 1302.1 to 6888.6 mg/100 g [13]. In light of this, reducing excessive sodium intake has emerged as one of the pivotal strategies outlined in the WHO’s global action plan for the prevention and control of noncommunicable diseases [1].

### 2.2. Salty Taste Perception Mechanism

Salty taste, as one of the five basic tastes in humans, primarily arises from the activation of taste receptors by NaCl and other mineral ions. It is particularly emphasized that taste receptors are specialized epithelial cells clustered in sensory end-organs; they can realize taste information processing, such as the chemical–electrical signal transformation in taste sensation, which is located primarily on the tongue and within the oral cavity that detects and responds to different taste stimuli [18,19,20,21]. It plays a key role in shaping food flavor and enriching people’s taste experiences. Moreover, salty taste perception also plays a vital role in regulating nutritional intake and electrolyte balance in the body, ensuring the ingestion of appropriate foods to maintain bodily health [22,23].

The mechanism of salty taste perception mainly involves taste receptor cells located on the surface of the tongue. When NaCl or other salty molecules bind to ion channels on taste receptor cells, it triggers a series of biochemical reactions [2,24]. At least two major conduction channels are currently known to participate in salty taste perception, namely the amiloride-sensitive and the amiloride-insensitive channels. The amiloride-sensitive channel mainly involves the epithelial sodium ion channel (ENaC), composing α-, β-, γ- and δ-subunits [19,25]. This class of channels exhibits high selectivity for sodium ions. When sodium ions enter the taste receptor cells through ENaC channels, they cause cell membrane depolarization, subsequently activating downstream signaling pathways, including the opening of CALHM1/3 ion channels and the release of ATP, eventually generating neural signals [20]. These signals are transmitted via taste nerves to the brain, where the taste cortex region deciphers them, resulting in salty taste perception [17,19,22,26].

Currently, research on the amiloride-insensitive channel, which is typically activated by high salt environments, remains unclear. The following outlines the candidate mechanisms currently under discussion. It perhaps includes transmembrane channel-like 4 (TMC-4, a novel Cl^−^ channel) and the TRPV 1 receptor associated with pain sensing [27,28,29]. Moreover, high concentrations of salt can activate bitter and sour taste receptor cells, which are innervated by neural pathways associated with aversion, eliciting a sensation of aversion to high-salt foods, thereby aiding in preventing excessive salt intake and protecting bodily health [22]. These findings suggest that salty taste perception via the amiloride-insensitive channel may depend on the collective stimulation of different taste cell groups, and also highlight the dual nature of salty taste, which only induces sensory pleasure when moderate [30]. Salty taste perception is a complex physiological process involving multiple taste receptors and signaling pathways. Understanding this mechanism aids in better comprehending the basis of taste perception. Although it is currently unclear how this knowledge will help develop strategies to reduce sodium intake, we will aim to explore this in future research.

### 2.3. Salty Taste Evaluation System

Evaluating saltiness intensity is an important part of salty taste research. By using sensory evaluation methods and high-precision instruments to measure the saltiness of food, we can understand its impact on consumer acceptance. As the bedrock of this system, sensory evaluation relies on human panelists to directly perceive and analyze the multidimensional sensory characteristics of food, including but not limited to salty taste, texture, aroma and appearance. Compared to other evaluation systems, sensory evaluation distinguishes itself through its intuitive and comprehensive nature. Techniques like Quantitative Descriptive Analysis (QDA) and Temporal Dominance of Sensations (TDS) enable evaluators to furnish detailed quantitative descriptions of food sensory attributes and dynamic sensory change information [31]. For instance, Duan et al. (2021) found that the addition of ginger could prolong the perception of salty taste in chicken soup through TDS evaluation and revealed the potential influence of ingredient combinations on the salty taste perception [32].

Nevertheless, sensory evaluation has certain limitations, including the subjectivity of evaluation outcomes and the likelihood of sensory fatigue among evaluators during prolonged evaluation sessions. In response, researchers have initiated exploration into objective evalution methods such as electronic tongue technology and cellular biology techniques, etc. (Figure 1). Electronic tongue technology measures the chemical properties of foods to predict or correlate with human taste perception. It employs multiple sensor arrays to perceive the overall characteristic response signals of samples, simulate sample identification, and conduct qualitative and quantitative analyses, providing rapid and consistent assessments related to salty taste. They offer advantages such as lower cost, higher throughput and the ability to safely assess the effects of ingredients not yet approved for human consumption or with limited toxicology data. However, while e-tongues can predict human taste perception, they require validation through human sensory methods and are not a substitute for human sensory testing. Therefore, e-tongues are valuable tools for screening and quality control but should not be considered the definitive standard for salty taste perception [33,34,35,36]. Cellular biology techniques, like constructing taste cell models, offer a novel perspective for assessing salty taste intensity by directly observing taste cell responses to salty substances of different strengths [20]. For example, by stimulating with NaCl, the taste receptor cell potential will increase in a dose-dependent manner within a certain range. This indicates that cell biology techniques have promising applications in the evaluation of saltiness intensity [37]. Facial expression analysis technology gauges consumer preferences for food saltiness by capturing facial movements. To some extent, it can reflect the degree of food preference, which can help research food recipes that are more popular with consumers (this benefit was consistent with the evaluation method for its saltiness intensity) [38,39]. Machine learning can amalgamate data from electronic tongues and other technologies to predict the salty taste intensity of food products and consumer acceptance [40].

In essence, sensory evaluation plays an important role in the evaluation of salty taste, but its objectivity and accuracy can be enhanced through integration with other technical modalities. Through the integration of multidimensional evaluation systems and data fusion strategies, researchers can provide a more accurate scientific basis for food formula improvement, process adjustment, quality evaluation and salty taste control to ensure the sensory quality of products and consumer satisfaction while reducing the sodium content of food. Moreover, it is essential to recognize that sensory assessments do not invariably predict consumer purchasing and eating behavior with complete accuracy, as this behavior is subject to a myriad of influences including brand, pricing, packaging and beyond [41]. This insight also presents us with a new perspective: incorporating consumer behavior more deeply into the sensory evaluation design process. Experiments will be crafted to emulate real-world consumption scenarios, employing techniques like home-use trials and virtual reality to replicate genuine consumer experiences, thus augmenting the applicability of sensory evaluation results [39,42].

### 2.4. Sodium Salt Reduction Strategies

Strategies for reducing sodium salt consumption are of paramount importance in global public health to mitigate the risk of chronic diseases associated with high-salt diets [43]. The WHO has set a target to reduce global sodium intake by 30% by 2025 [4]. Various national governments and organizations have implemented diverse measures to attain this objective. In the United Kingdom, the government has implemented several policies, including public awareness campaigns, improvements in food salt content labeling and adjustments in food formulation to decrease sodium levels [13,44,45]. South Africa has enacted legislation to restrict the maximum sodium content in processed foods, and Finland has enforced sodium content labeling regulations [17]. Additionally, China has implemented interventions such as distributing salt spoons, promoting awareness of high-sodium foods, and highlighting salt content on food packaging [13,46].

These measures have yielded positive outcomes. For instance, the UK has successfully reduced salt content in over 80 foods by 16%, while stroke prevalence has also declined in Finland and Poland by approximately 10.1% and 23.1%, respectively. In Japan, per capita salt intake decreased from 14.5 g in 1973 to 9.5 g in 2017, and China has witnessed a gradual decline in per capita cooking salt consumption over the years (averaging a 2 g/d decrease per decade). Nevertheless, current salt intake levels surpass the WHO’s recommended criteria. Consequently, the ongoing challenge in salt reduction efforts lies in further diminishing sodium content in food products while preserving their quality and sensory appeal. Researchers are exploring diverse methods and techniques to achieve this, including the use of salt substitutes, altering the shape and structure of salt particles, innovative food processing technologies (e.g., ultrasound, high-pressure processing, etc.) and leveraging multisensory synergistic effects to enhance salty taste perception (Table 1).

#### 2.4.1. Salt Substitutes

In salt reduction strategies, the utilization of salt substitutes emerges as an effective approach. Typically, these substitutes consist of metal salts with minimal or no NaCl content, such as potassium chloride, magnesium chloride, and calcium chloride [14]. These salts partially mimic the salty taste and preservative effects of NaCl. Among them, potassium chloride stands out as one of the most established and widely employed salt substitutes [1]. Not only does it impart a salty taste and antibacterial properties akin to NaCl, but it also helps maintain the water content and texture of myofibrillar gel. Furthermore, potassium plays a crucial role in balancing sodium ions in bodily fluids, thereby reducing the risk of stroke and coronary heart disease [47]. Studies have indicated that moderate substitution of potassium chloride for NaCl in cheese preserves the flavor and microbial quality. However, excessive replacement ratios may result in metallic and bitter tastes, adversely affecting overall palatability [1,2,22,48,49]. In addition, magnesium chloride and calcium chloride can serve as partial substitutes for NaCl. Nonetheless, their low salinity and limited permeability in preserved foods pose challenges in achieving desirable flavor profiles.

To optimize substitution effects, researchers have experimented with combining multiple metal salts in specific proportions to reduce NaCl content without compromising food quality and consumer acceptance [26,50]. For instance, in sausage production, replacing 40% of NaCl with a mixture of potassium lactate and calcium ascorbate did not significantly diminish overall consumer acceptance [1]. However, it is essential to recognize that salt substitutes may not be suitable for all food products. For example, the application of calcium carbonate and calcium chloride in bread could lead to notable alterations in sensory quality [16]. Moreover, excessive intake of certain salt substitutes may pose health risks. For instance, overconsumption of potassium-containing substitutes may elevate the risk of hyperkalemia [17,41]. Hence, when developing and applying salt substitutes, it is imperative to consider the specific requirements of food products and prioritize consumer health and safety. This ensures the efficacy and safety of salt reduction strategies [4,48].

#### 2.4.2. Modification of the Shape and Structure of Salt

Altering the shape and structure of salt represents a viable strategy to achieve salt reduction objectives. Salty taste perception is intricately linked to the dissolution and diffusion rates of salt in the oral cavity. Decreasing salt particle size and increasing specific surface area can expedite salt dissolution and diffusion in saliva, thereby enhancing the transmission efficiency of sodium ions to taste buds and intensifying salty taste perception [1,49,51].

Traditional cubic or octahedral-shaped salt crystals may result in inadequate sensing of sodium ions being ingested. However, utilizing salt with unique morphological structures such as nano salt, sheet salt, and hollow salt microspheres can heighten perceived salty taste intensity without altering the quality and sensory experience of foods like pasta and meat products [21,26]. Nano salt, characterized by its small particle size (approximately 20 μm), swiftly disperses into the food matrix, delivering a stronger salty taste [1]. Plate salts boast increased solubility and diffusion rates due to their larger surface area [26]. Hollow salt microspheres reduce actual sodium content through their internal hollow structure while maintaining consistent salty taste perception. Nonetheless, these alterations in salt structure and morphology may lead to a rapid decline in salty taste perception, presenting challenges for their application in liquid products [1]. Furthermore, the high production cost serves as a limiting factor for widespread adoption. Continuous optimization of preparation processes and cost reduction efforts are anticipated to enhance the applicability of salt reduction strategies involving shape and structure modification in the future.

#### 2.4.3. Innovation of Food Processing Technology

In salt reduction strategies, innovation in food processing technology plays a crucial role in enhancing the sensory quality and preserving low-salt foods. High-pressure and ultrasonic techniques, as primary non-thermal processing methods, exhibit significant potential in low-salt food processing [43,52].

High-pressure technology, applying pressures ranging from 100 to 1000 MPa, effectively inhibits microbial growth and improves color, structural quality and water retention in low-salt foods. Furthermore, it enhances the perception of salty taste by reducing sodium ion interactions with proteins and facilitating sodium ion release. For instance, treatment at 200 MPa maintains physicochemical and sensory characteristics in low-salt beef sausages comparable to normal-salt beef sausages, while ham treated with high pressure instead of rolling processes exhibits heightened salinity [1]. Ultrasonic technology employs cavitation and mechanical effects to alter salt distribution in food matrices, thereby improving the perception of salty taste and overall acceptance of low-salt foods. Ultrasound also accelerates the diffusion rate of table salt and inhibits microbial growth by altering cell membrane permeability. Notably, ultrasonic treatment significantly enhances the flavor, saltiness, texture and overall acceptance of low-sodium ham [26,53,54]. Despite demonstrating potential in salt reduction, these technologies face challenges such as high costs and the need for application transformation that must be addressed.

## 3. Detailed Multisensory Synergistic Effect

In the quest for salt reduction strategies, despite advancements like salt substitutes, alterations in salt morphology, and innovations in food processing technology, solutions are somewhat hampered by the resulting unpleasant taste and cost implications. Consequently, their broad implementation in food production is hindered. Food flavor is the result of a combination of various sensory systems, encompassing taste, smell, auditory cues, visual cues, tactile sensations and trigeminal nerve perception. Leveraging the interplay among these senses for enhanced salt perception via cross-modal sensory integration emerges as a promising and innovative avenue for salt reduction (Figure 2) [55].

### 3.1. Enhancement of Salty Taste Perception through Taste Interaction

When two or more taste substances interact in the oral cavity, their combination can alter taste perception, a phenomenon termed intertaste interactions. These interactions may lead to taste enhancement, inhibition or even the emergence of new taste perceptions [2,22,26,56]. In investigating the impact of taste interaction on salty taste perception, it has been observed that specific concentrations of umami and sour substances can augment salty taste perception by activating taste buds, increasing saliva secretion, promoting food dissolution, facilitating sodium ion dissociation and enhancing contact with membrane ion channels [57].

Umami is the taste response to substances such as amino acids and nucleotides, and it converges with salty processing in the central nervous system [22]. Through chemical interactions, oral physiological processes, and cognitive neural effects, the combination of umami and salty tastes effectively heightens salty taste perception [26,58,59,60]. The enhancing effect of umami substances on salty taste is influenced by salt concentration, umami substance type and its strength [61]. Moreover, synergies between umami substances further intensify salty taste, such as the combination of sodium inosinate and MSG, which yields a stronger salty effect compared to individual usage [59].

Also, sour substances exhibit a dose-dependent dual effect on salty taste perception. Across a range of concentrations, sour taste substances like citric acid can enhance salty taste perception. However, with increasing concentrations, this enhancing effect may shift towards inhibition [5]. The interaction between sour and salty tastes not only directly enhances salty taste perception but also enriches the taste profile of food, compensating for sensory declines resulting from reduced salt usage [2,53,60]. These taste interactions positively impact consumer acceptance and preference for low-salt foods, thereby promoting their consumption. Thus, through judicious use of umami and sour taste substance interactions, it is possible to effectively enhance salty taste perception and enrich the overall flavor experience of food.

### 3.2. Enhancement of Salty Taste Perception through Olfactory Interaction

Olfaction plays a pivotal role in food flavor perception, with reports suggesting that up to 80% of perceived food flavor is contributed by olfactory receptors, and that the taste of food can be readily influenced by its aroma [35]. Olfactory perception unfolds in two stages: anterior nasal cavity perception and posterior nasal cavity perception. Volatile aroma compounds are inhaled through the nasal cavity, binding with olfactory epithelial cells to generate specific electrical signals. These signals are then encoded by olfactory bulb cells and processed by the cerebral cortex, leading to anterior nasal cavity aroma perception. Subsequently, posterior nasal cavity aroma perception occurs when aroma compounds released during chewing reach the olfactory epithelial cells via oral respiratory airflow. This perception can bypass taste nerves, directly activating the taste cortex and eliciting an interaction with taste perception [62]. Studies indicate that the simultaneous stimulation of taste and olfactory receptors activates brain regions (such as the orbitofrontal cortex, OFC) associated with both receptor types, resulting in overlapping activation and heightened taste perception [26]. Odor compounds with salty flavor characteristics, aligned with the taste of salt, notably enhance the perception of saltiness in the human body. This phenomenon, termed odor-induced salty taste enhancement (OISE), represents an effective strategy for enhancing the perception of salty taste in low-salt foods [5,41,63,64,65]. Multiple studies have reported the potential of aroma substances to enhance the flavor of low-salt foods [53,66].

Numerous aroma compounds with the ability to enhance salty taste have been identified in sauces, cheese, meat products (bacon, ham, beef), seafood (sardines) and herbs [65,67,68]. However, the type, intensity, combination of aroma compounds, and salt solution concentration can all influence the effect of OISE [51,62,67]. For example, while the addition of ham flavor enhances saltiness in cheese, tomato and carrot flavors do not contribute to salty strength [46,63]. The effect of OISE of 3-methylpropionaldehyde found in soy sauce was highest in a 0.3% salt solution and diminished with a 0.8% salt concentration [46]. Furthermore, cooking methods and oral processing can impact the interaction between odor and salty taste. Varied cooking treatments yield different aroma compounds, while oral processing, such as saliva’s role, influences migration and release [63,68]. At the neural level, the integration of fragrance and salty taste occurs in cognitive brain regions like the OFC, anterior cingulate cortex and temporal lobe [35,69]. Odor-induced taste enhancement is associated with increased activity in cognitive brain regions. Individuals can recall the smell of salty food, and upon encountering familiar salty aroma components, can experience induced taste through associative functions [65]. Hence, through careful selection and addition of aroma substances aligned with salty taste, it is possible to maintain or even enhance the perception of salty taste in food while reducing salt content, thereby promoting the adoption of healthier dietary habits [70,71].

### 3.3. Enhancement of Salty Taste Perception through Auditory Interaction

Hearing serves as a crucial element of the multisensory experience, uniquely regulating the perception of food [72]. It not only enriches our dining encounters but also influences our perception of salty taste through cross-modal interactions.

During eating, the sounds produced by chewing and swallowing directly impact our perception of food’s saltiness. Research indicates that food consumption sounds, such as the crisp crunch of potato chips, can heighten consumers’ perception of salty taste, thereby enhancing the overall flavor experience [73]. Music, as a non-invasive sensory stimulus, can also modulate the perception of salty taste by altering our eating rhythm and emotional state. Pleasant background music can foster a positive dining ambiance, potentially slowing down chewing speed and prolonging food residence time in the mouth, thereby intensifying the release of salty substances and sodium ions, thus enhancing salty perception [74]. Furthermore, music can influence consumers’ food choice behavior, encouraging them to opt for low-salt options, thus promoting healthy dietary habits [73]. In contrast, the presence of noise may disrupt salty taste perception [41]. Specific tones in noise can activate neurons involved in odor response, directly impacting taste perception by influencing odor transduction. Therefore, creating a quiet dining environment may enhance food’s salty perception and overall eating experience [72,75].

People’s cognitive abilities enable them to associate specific sounds with taste experiences, influencing food’s saltiness perception. For instance, the aroma of soy sauce is often linked to salty taste, and auditory stimuli can similarly shape taste experiences through cognitive pathways [41]. In an experiment, participants accurately matched specially designed soundtracks with a salty taste, suggesting that sounds can modify individual food and drink taste experiences by creating taste expectations, directing attention or influencing thought and perceptual processes [73]. This cognitive research application can extend to sound seasoning, emphasizing or enhancing food flavor properties through music with specific sound characteristics. Research indicates that music associated with salty taste often features irregular sound attributes, including rhythmic patterns such as bass frequency, long decay time, and high auditory roughness [76].

In summary, hearing interacts with taste in various ways to enhance salty taste perception. When designing food and food environments, considering sound factors is essential to improve the acceptance of low-salt foods and promote healthy eating habits. In the future, sound could emerge as an innovative flavoring tool to help reduce dietary salt intake and advance public health.

### 3.4. Enhancement of Salty Taste Perception through Visual Interaction

Vision is a pivotal component of the multisensory experience, playing a critical role in the perception of food. Through cross-modal correspondence, individuals can associate specific visual attributes with taste qualities, thereby influencing the perception of salty taste in food [77,78]. Food color, as a prominent visual cue, profoundly impacts sensory properties by interacting with receptors in the retina through reflected light or emitted visible spectrum, thereby shaping our perception of food and dietary preferences [35]. For instance, the color of food, packaging, accompaniments and tableware can influence the assessment of taste and food quality, subsequently impacting dietary preferences and food choices. Usually, serving salty dishes on black plates or pairing brown steak with vibrant green and red vegetables not only enhances visual appeal but also heightens consumers’ perception of salty taste [41,79,80,81].

Moreover, the appearance and shape of food can also influence taste perception. An emotion-mediated association exists between taste and shape, suggesting that visual attributes such as food shape, packaging design, and environmental geometry are linked to specific tastes [78,80]. For instance, chips with a coarse appearance may evoke a saltier sensation [41]. Leveraging the learned association between vision and taste, designers can consider these factors when crafting food appearance to augment consumer perception of salty taste [44,77,82].

Similarly, environmental lighting not only impacts visual perception but also influences taste perception through sensory compensation mechanisms. Diminished lighting diminishes visual input while enhancing the perception of taste and smell. Consequently, consumers may exhibit increased sensitivity to the salty taste of food in such settings, thereby elevating the overall dining experience. For instance, some restaurants opt for dimly lit environments to intensify the flavors of dishes, with tomato soup perceived as saltier under subdued lighting compared to bright illumination; similarly, consuming potato chips in a darkened cinema may enhance the perception of saltiness [83].

In summary, vision plays a significant role in multisensory synergistic strategies for salt reduction. By thoughtfully orchestrating the color, appearance, shape of food and environmental lighting, we can heighten the perception of salty taste without augmenting salt content, thereby enriching the flavor experience of low-sodium foods. This approach not only aids in reducing salt intake but also fosters the cultivation of healthy eating habits, thereby contributing to public health [74]. Future research and practices should delve deeper into the potential of vision-taste interactions, offering novel insights and methodologies for the development and promotion of salt-reduced foods.

### 3.5. Enhancement of Salty Taste Perception through Tactile Interaction

The tactile perception of the human body to food occurs during the processes of chewing and swallowing in the mouth, as well as in the tongue, pharynx, esophagus, and through hand touch. These tactile sensations encompass properties such as hardness, roughness, temperature and texture of food, which are closely intertwined with emotions and pleasure, influencing the taste experience [84]. Oral touch serves as the primary mode of food perception [21]. Research indicates that taste receptors are likely to exhibit sensitivity to tactile stimuli, particularly within oral tissues like the tongue, hard palate and gingiva [18,20,21]. These regions possess tactile sensitivity to food surface roughness and shape, facilitating enhanced contact between salty substances and taste buds, thereby augmenting salty taste perception. For instance, the process of chewing food into smaller particles within the mouth, combined with saliva mixing, not only aids in food dissolution but also expands the contact area for salty substances with taste receptors, optimizing the delivery of salty stimuli to ion channels [85]. Active oral processing during food contact is suggested to create a more conducive chemical environment for taste receptor cells, enhancing taste sensation. Research findings indicate that the delicious taste perceived during active tasting is 3.8 times stronger than during passive tasting due to intensified oral cavity movements and increased tactile stimulation [86].

Moreover, food texture, including aspects like roughness and irregularity, significantly influences salty taste perception. The tactile experience is closely associated with food roughness, as a rough food surface can elicit more intense tactile stimulation, thus enhancing salty taste perception [84]. For instance, potato chips with a rough surface texture can evoke stronger tactile sensations compared to smoother chips, consequently heightening sensitivity to salty taste. Irregular salt particles can also interact with the human body tactilely during oral processing, further enhancing salty taste perception. Prolonged exposure to certain food textures leads to associative learning between salty taste and tactile sensations. Individuals may associate salty taste with food roughness and irregularity, a cognitive link that can be exploited through food design [76]. By investigating the cross-modal correspondence between taste and touch, and applying these findings to salty taste perception and the development of low-salt foods, it is possible to better guide consumers towards choosing reduced-sodium options [44].

### 3.6. Enhancement of Salty Taste Perception through Trigeminal Interaction

Trigeminal sensory sensation, as a vital component of the somatosensory system, plays a crucial role in food perception. The trigeminal nerve (V cranial), through its sensory endings, responds to temperature, pressure, pain and other stimuli, thus serving as a mechanism for pain and warning in mammals. Trigeminal nerve sensation encompasses various sensations, including burning, tingling, warmth, cooling and trembling [87], all of which significantly influence flavor perception, particularly salty taste perception [88,89].

Numerous foods contain chemical stimuli that enhance salty taste perception, such as capsaicin in cayenne pepper, piperine in black pepper, isothiocyanate esters in mustard, cassia bark phenol in herbal medicine and carbon dioxide in carbonated drinks. These stimuli interact with oral-specific receptors (such as TRPV 1, TRP channels), resulting in trigeminal sensory responses. The interaction between trigeminal sensation and salty taste perception involves various aspects, including taste perception mechanisms, dose effects and neurotransmission [89,90,91]. For example, if the concentration of trigeminal stimuli such as capsaicin is too high, the heat and pain induced by spiciness can overshadow the perception of saltiness [90,91,92]. Additionally, upon stimulation of the oral cavity by foreign substances, the human body integrates signals and may release endorphins, thereby heightening sensitivity to taste and odor and enhancing salty taste perception. For instance, capsaicin was found to enhance responses in the cerebral cortex and OFC to high-salt stimulation, indicating the central role of the trigeminal nerve in salty taste perception [93,94].

However, individual differences exist in this interaction, influenced by factors such as age, gender, and associative learning between food and salty taste. For example, Zhang et al. (2020) conducted a study examining the impact of the pungent sensation induced by Sichuan pepper oleoresin on the sensory experience of saltiness in both younger and older populations. Their findings indicate that younger individuals are more sensitive to salty and spicy tastes, and have lower recognition thresholds than older individuals. The study also discovered a gender difference, with men exhibiting lower recognition thresholds for spicy tastes than women. However, the authors emphasize that these findings are preliminary and require additional research to fully comprehend the complex interactions at play [95]. Besides, associative learning also contributes to the increased perception of saltiness in spicy foods, as salty taste is frequently associated with spicy dishes. Future research and development in the food industry can further explore the role of the trigeminal nerve in salty taste perception and devise healthier and more delicious salt-reducing foods.

### 3.7. Enhancement of Salty Taste Perception through Multisensory Synergistic Interaction

In our investigation of strategies for multisensory synergistic salt reduction, we acknowledge the complexity of food perception, which involves multiple sensory modalities. These modalities encompass taste, smell, touch, vision and trigeminal perception, all of which interact to shape our perception of salty foods (Figure 2) [22].

Within multiple sensory systems, diverse sensory interactions can lead to complex perceptual outcomes. In the cross-modal odor-flavor-flavor ternary system, such as odor-sour-salty mixtures, sour taste not only directly enhances salty taste, but also amplifies salty taste perception by boosting salt-related odor perception [5]. Similarly, the combination of MSG and cheese odor with NaCl has been found to enhance salty taste perception more than when each component is used alone [63,64,96]. Furthermore, the integration of trigeminal nerve sensation, triggered by stimuli like capsaicin, with taste and aroma sensation in the OFC of the brain heightens sensitivity to taste and odor, thereby enhancing salty taste perception in complex food substrates [87,93]. In the color-odor-taste model, color can influence salty taste perception by impacting odor perception. Odor recognition diminishes when odors lack color cues or are paired with inappropriate colors, resulting in weakened odor-induced salty enhancement [97]. Additionally, food texture and composition can influence this cross-modal interaction; for example, sardine fragrance in low-fat and low-salt foods can significantly heighten salty taste perception [26,69].

The brain plays a central role in multisensory synergistic salt reduction. Top-down cognitive modulation, mediated by higher-order brain activation, is crucial for the multisensory-induced enhancement of salty taste perception. This enhancement is closely tied to individuals’ learning, repeated exposure and associative processes, and the interaction of different senses can modulate salty taste perception. Moreover, this multisensory synergy can impact flavor perception, eating expectations, and behavior, enabling consumers to enjoy a pleasant eating experience with low-salt foods [35,64,98].

## 4. Conclusions and Outlook

Table salt is an indispensable component of food, and controlling its intake is crucial for public health. Despite various salt reduction strategies such as salt substitutes, modifications in salt structure and innovations in food processing technology, they encounter challenges related to introducing undesirable flavors and high costs. Leveraging cross-modal interactions among taste, smell, hearing, vision, touch and trigeminal perception can effectively reduce salt intake without sacrificing food flavor. However, current research on multi-sensory synergy for salt reduction primarily focuses on simplistic model systems, neglecting complex food systems. Additionally, while recent studies have deepened our understanding of salty taste perception mechanisms and improved salty taste evaluation systems, the intricate interaction mechanisms among multiple senses remain incompletely elucidated. Future research endeavors aim to integrate modern computer technology and cognitive neuroscience to thoroughly clarify mechanisms underlying salty taste perception and multi-sensory synergy for salt reduction by fully understanding the complexity of food matrices and human sensory organs, precisely constructing suitable experimental matrices, and applying multisensory synergy for salt reduction to practical food processing. Furthermore, it is crucial to combine multi-sensory synergy for salt reduction with salt awareness education. Through collaborative efforts from governments, organizations, and other stakeholders, there is a need to advocate for reducing daily salt intake among the population to improve public health.

## Figures and Tables

**Figure 1 foods-13-01659-f001:**
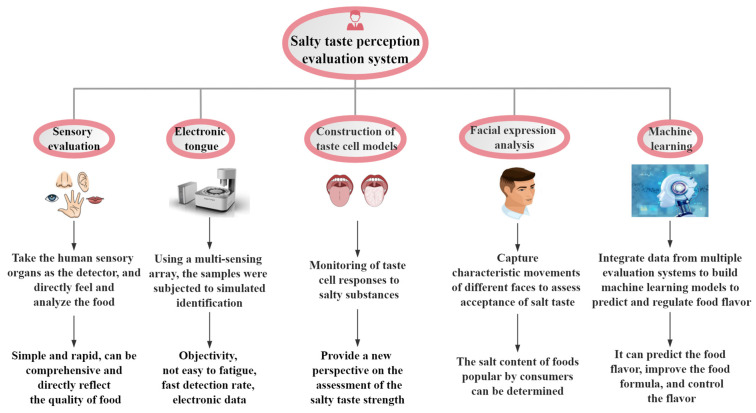
Various salty taste perception evaluation system.

**Figure 2 foods-13-01659-f002:**
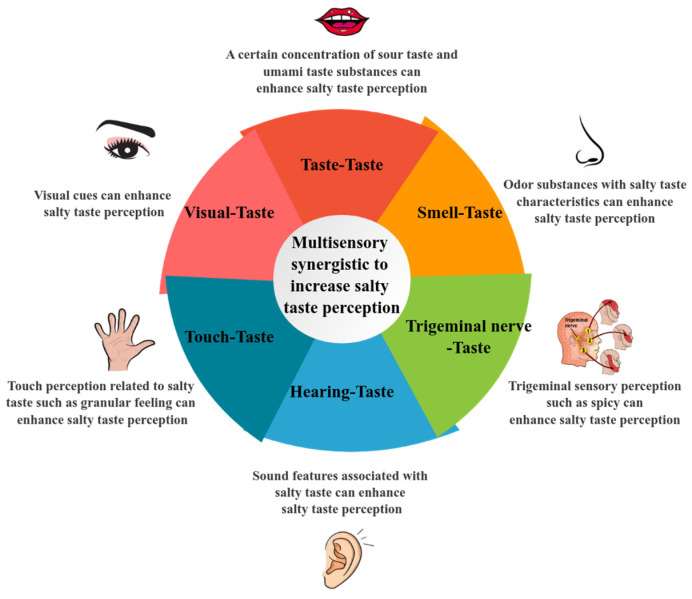
Multisensory synergistic effect to enhance salty taste perception.

**Table 1 foods-13-01659-t001:** Advantages and disadvantages of different salt reduction strategies.

Salt Reduction Strategy Category	Introduction to Salt Reduction Strategy	Advantage	Disadvantage	Improved Method	References
Salt substitutes	Replacing partial NaCl with metal salts	Suitable for most food, put into food production already	Excessive addition can introduce a bad flavor	Various metal salts are prepared in a certain proportion	[1,15,42,43]
Changing the shape and structure of salt	Change the particle size, specific surface area of the salt, etc.	No adverse effects on food quality	High preparation cost and difficult application	Optimize the preparation technique	[19,46,47]
Food processing technology innovation	High pressure, ultrasonic technology, etc.	Improves the decline of sensory texture of food caused by salt reduction	The cost of technology development is high	For the study of new salt-reducing foods	[37]
Multisensory synergistic effect	Using the interaction of multiple senses	Excellent effect in taste, color, aroma, acceptability	The interaction mechanism needs further investigation	The interaction mechanism is expounded from multiple levels	[36]

## Data Availability

The original contributions presented in the study are included in the article, further inquiries can be directed to the corresponding author.

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
