# Peer review of "Progress in Multisensory Synergistic Salt Reduction"

_foods, 2024, doi:10.3390/foods13111659_

Round 1

Reviewer 1 Report

Comments and Suggestions for Authors

plz see the attached file.

Reviewer 2 Report

Comments and Suggestions for Authors

The manuscript provides a description of multisensory aspects related to human salt consumption. It is well written and can be considered for publication. However, it still contains some flaws that must be corrected before further processing. Please check them as follows:

Abstract – the authors write that a technical guidance to create food alternatives will be provided with this revision. However, in the review they describe mostly sensory aspects of salt consume. That should be revised.

Minor correctios should be provided:

Line 102 - double hyphen for “channel--like 4”?

Line 102 - negative charge seems not to be superscripted. Please check.

Line 112 - please add space in the beginning of sentences.

Line 133 - please add a proper citation with year of publications. Please also add the number of the citation.

Line 156 - Please begin sentences with capitalized words.

Table 1 - text alignment seems not to be standardized in Table.

Table 1 - please check space between columns.

Figure 2 - please remove the board lines in the small figures.

Figure 2 - please proper align the texts around the figure.

Reviewer 3 Report

Comments and Suggestions for Authors

The article reviews methods to reduce salt intake without compromising food flavor by utilizing multisensory synergies, highlighting the potential and challenges of these strategies in promoting public health.

Reviewer's comments:

- Lines 30-32: Please provide more specific context or evidence to justify the claim that salt is the "cornerstone" of the modern food industry and dietary culture, as well as the "king of all flavors."

- Line 36-33: Please provide more specific information or data to support the statement that salt intake is far beyond physiological needs and elaborate on the link between high salt intake and health risks such as hypertension, heart disease, and stroke.

- Line 67-70: Please provide more specific data for different regions or populations to support this claim.

- Line 286-289: Please provide direct evidence or studies to support the claim that multisensory synergy can alleviate conditions like anorexia and obesity.

Comments on the Quality of English Language

Minor editing of English language required.
